# Biomarkers of Oxidative Stress in Diabetes Mellitus with Diabetic Nephropathy Complications

**DOI:** 10.3390/ijms241713541

**Published:** 2023-08-31

**Authors:** Petya Goycheva, Kamelia Petkova-Parlapanska, Ekaterina Georgieva, Yanka Karamalakova, Galina Nikolova

**Affiliations:** 1Propaedeutic of Internal Diseases Department, Medical Faculty, Trakia University Hospital, 6000 Stara Zagora, Bulgaria; petya.goycheva@trakia-uni.bg; 2Medical Chemistry and Biochemistry Department, Medical Faculty, Trakia University, 11 Armeiska Str., 6000 Stara Zagora, Bulgaria; kamelia.parlapanska@trakia-uni.bg (K.P.-P.); ekaterina.georgieva@trakia-uni.bg (E.G.); 3Department of “General and Clinical Pathology, Forensic Medicine, Deontology and Dermatovenerology”, Medical Faculty, Trakia University, 11 Armeiska Str., 6000 Stara Zagora, Bulgaria

**Keywords:** diabetes mellitus, diabetic nephropathy, oxidative stress, NO radicals, NOS

## Abstract

The present study aimed to investigate and compare biomarkers of oxidative stress and the activity of antioxidant enzymes in the plasma of patients with different stages of diabetic nephropathy. For this purpose, we studied (1) the levels of reactive oxygen species and reactive nitrogen species as oxidative stress parameters, (2) lipid and protein oxidation, (3) the activity of antioxidant enzymes, and (4) cytokine production. Patients with type 2 diabetes mellitus were divided into three groups according to the loss of renal function: patients with compensated diabetes mellitus with normal renal function DMT2N0 measured as an estimated glomerular filtration rate (eGFR) ≥ 90 mL/min/1.73 m^2^, a group with decompensated diabetes mellitus with complication diabetic nephropathy and mild-to-moderate loss of renal function DMT2N1 (eGFR < 60 mL/min/1.73 m^2^: 59–45 mL/min/1.73 m^2^), and a decompensated diabetes mellitus with diabetic nephropathy group with moderate-to-severe loss of renal function DMT2N2 (eGFR > 30 mL/min/1.73 m^2^: 30–44 mL/min/1.73 m^2^). All results were compared with healthy volunteers. The results showed that patients with diabetic nephropathy had significantly higher levels of ROS, cytokine production, and end products of lipid and protein oxidation compared to healthy volunteers. Furthermore, patients with diabetic nephropathy had depleted levels of nitric oxide (NO), an impaired NO synthase (NOS) system, and reduced antioxidant enzyme activity (*p* < 0.05). These findings suggest that patients with impaired renal function are unable to compensate for oxidative stress. The decreased levels of NO radicals in patients with advanced renal complications may be attributed to damage NO availability in plasma. The study highlights the compromised oxidative status as a contributing factor to impaired renal function in patients with decompensated type 2 diabetes mellitus. The findings of this study have implications for understanding the pathogenesis of diabetic nephropathy and the role of oxidative stress and chronic inflammation in its development. The assessment of oxidative stress levels and inflammatory biomarkers may aid in the early detection and prediction of diabetic complications.

## 1. Introduction

Diabetic nephropathy (DN) is a common complication affecting more than 40% of patients with type 1 and type 2 diabetes. DN is a chronic and progressive kidney disease and can remain clinically silent throughout a person’s life [1,2,3]. The development of diabetic nephropathy is attributed to long-term exposure to high blood glucose levels, which can negatively impact beta-cell function. The high glucose levels might contribute to free radical generation and development of oxidative stress (OS). Disturbed redox balance and OS in the body further worsen the glucose regulation seen in type 2 diabetes [4]. Oxidative stress, inflammation, and associated endothelial dysfunction play a significant role in the development and progression of diabetic nephropathy. The primary pathological mechanism linking OS, inflammation, and diabetic nephropathy involves the initial kidney damage caused by reactive oxygen species (ROS) and reactive nitrogen species (RNS) [5,6]. Moreover, high levels of ROS/RNS can cause damaging modifications to lipids, proteins, and DNA. These oxidation products have a longer lifespan and can serve as markers to assess the redox status of the body. An understanding of the role of oxidative damage is crucial for the management and prevention of diabetic complications. The use of strategies to mitigate the oxidative damage effects of diabetes-related complications may make it possible to reduce the number of patients at risk [7].

At physiological levels, free radicals and their metabolites play an essential role in the biomodulation and regulation of processes, such as signal transduction, gene expression, and various epigenetic modifications, by modifying the activity of metabolic enzymes and transcription factors [8]. These averages are defined as short-term and transient adaptations of biological systems that occur in response to exposure to sub-toxic concentrations of ROS and RNS, followed by their efficient removal [9]. According to the intensity, oxidative stress can be classified as physiological oxidative stress (eustress) with low or moderate levels of oxidants and distress, expressed in toxic oxidative load due to high exposure to free radicals, causing damage to many biomolecules [10].

Prolonged exposure to high glucose levels accompanied by OS leads to damage and structural changes in the renal nephron, particularly at the glomerulus level. These structural changes in the glomerulus contribute to the progressive decline in kidney function seen in diabetic nephropathy [11]. The glomerular filtration barrier (GFB) is a crucial component of kidney function, and it includes the mesangium, glomerular basement membrane (GBM), fenestrated glomerular endothelial cells, and podocytes [4]. In DN, there are pathological changes that occur within the GFB, such as thickening of the GBM, mesangial sclerosis (scarring of the mesangium), and dysfunction of the glomerular endothelial cells, which can lead to damage to the glycocalyx [5]. Understanding the mechanism of pathology is essential for developing strategies to prevent and manage this complication of diabetes. Early detection can help slow down the progression of diabetic nephropathy and reduce the risk of kidney failure [7].

The article is focused on oxidative status examination and antioxidant defense mechanisms in diabetic nephropathy patients with different stages of kidney damage. The investigation included the measurement of ROS and RNS levels as indicators of OS, assessment of lipid and protein modification, the evaluation of the antioxidant enzyme activities, and pro-inflammatory cytokine analysis, which contribute to the progression of diabetic nephropathy (DN). The presented results could provide valuable information regarding the underlying mechanisms of DN and mark potential therapeutic strategies to mitigate the condition’s progression.

## 2. Results

### 2.1. Plasma Levels of NO, eNOS and iNOS

The DMT2N0 group with normal kidney function measured as an estimated glomerular filtration rate (eGFR) ≥ 90 mL/min/1.73 m^2^ (stage 1) showed significantly higher levels of nitric oxide radicals (NO•) compared to the control (*p* = 0.00014) (Figure 1A) and the stage 3a (*p* = 0.0002) and stage 3b (*p* = 0.00024) DN groups. The eNOS (Figure 1B) and iNOS (Figure 1C) concentrations were significantly lower in the groups with renal function loss compared to the group with normal kidney function (stage 1).

### 2.2. Assessment of Oxidative Stress by Measuring the MDA, ROS and 8-Iso-prostaglandin Levels

The DN groups, both with mild-to-moderate loss of renal function (eGFR stage 3a; *p* = 0.00) and moderate-to-severe loss (stage 3b; *p* = 0.0002), showed similar values and significantly higher levels of malondialdehyde (MDA) compared to the DMT2N0 group with normal renal function (eGFR stage 1). A statistically significant increase in the levels of MDA (Figure 2A) in all three groups of patients compared to healthy volunteers (*p* = 0.00014) was detected.

ROS levels (Figure 2B) and plasma 8-iso-prostaglandin (8-Iso-PGF) concentration (Figure 2C) were also increased in the DN groups compared to the DMT2N0 group with normal renal function.

The correlation study revealed a significant negative study between the plasma MDA and eGFR (*r* = −0.88; *p* ≤ 0.0001; Figure 3A). A positive association is observed between plasma MDA and UAE (*r* = 0.80; *p* ≤ 0.00013; Figure 3C). Similarly, plasma 8-IsoPGF depicted a significant negative relation with eGFR (*r* = −0.98; *p* ≤ 0.00014; Figure 3B), and a positive association of 8-IsoPGF and UAE (*r* = 0.84; *p* ≤ 0.00014; Figure 3D).

### 2.3. Levels of Protein Carbonyl and Advanced Glycation End Products, as Suitable Surrogate Biomarkers of Oxidative Damage in Plasma Samples Diabetic Patients

Compared with controls, protein carbonyl (PC) levels (Figure 4A) were statistically significantly higher in all groups of patients with and without complications: DMT2N0 stage 1 (*p* = 0.0001), DMT2N1 stage 3a (*p* = 0.00014), and DMT2N2 stage 3b (*p* = 0.00015). In the groups of diabetic patients with impaired renal function, the PC values were close and statistically insignificantly higher compared to DMT2N0 stage 1 (*p* = 0.0001).

In the DMT2N0 (Figure 4B), AGEs levels showed statistically significant elevation compared to controls (*p* = 0.0001). In the groups with impaired renal function, the AGE levels were statistically significantly increased compared to the control group (DMT2N1 stage 3a, *p* = 0.0001, and DMT2N2 stage 3b, *p* = 0.0003). The same dependence was observed in the comparison between DMT2N0 and DMT2N1 stage 3a (*p* = 0.0001) and DMT2N2 stage 3b (*p* = 0.0002).

The correlation study revealed a significant negative study between the plasma PC and IL-10 (r = −0.97; *p* ≤ 0.0002; Figure 5A) and a positive association is observed between plasma AGEs and IL-10 (r = −0.97; *p* ≤ 0.0001; Figure 5B). 

### 2.4. Antioxidant Enzymes’ Activities (SOD, CAT and GPx) in Patients with DN

The results analysis indicated that, in patients with DN, the activity of antioxidant enzymes, including superoxide dismutase (SOD; Figure 6A), catalase (CAT; Figure 6B), and glutathione peroxidase (GPx; Figure 6C) was significantly reduced. 

The activity of the antioxidant system in DMT2N1 stage 3a and DMT2N1 stage 3b, was significantly depleted. Activity of SOD in DMT2N1 stage 3a was statistically significantly decreased compared to controls (DMT2N1, stage 3a; *p* = 0.0001) and compared to DMT2N0 stage 1 (*p* = 0.0002). The SOD activity in DMT2N1 stage 3b showed similar results. 

The results indicated a statistically significant difference in CAT activity between the diabetic nephropathy groups, DMT2N1 (*p* = 0.000128) and DMT2N2 (*p* = 0.00013), and diabetic patients without complications, DMT2N0 (Figure 3B). The CAT activity was significantly higher in DMT2N1 (*p* = 0.0001) and DMT2N2 (*p* = 0.00012) compared to controls. 

The analysis of results indicated that both groups of patients with impaired renal function, DMT2N1 (0.00014) and DMT2N2 (0.00011), had significantly lower GPx activity (Figure 3) compared to DMT2N0 and controls (*p* = 0.0001).

### 2.5. The Levels of Pro-Inflammatory Cytokines IL-6, TNF-α, IFN-γ

In both groups of patients with DMT2N1 and DMT2N2 with loss of renal function, interleukin 6 (IL-6) levels (Figure 7A) were statistically significantly increased compared to both DMT2N0 with normal renal function (*p* < 0.05) and the controls (*p* < 0.05). The mean value for the tumor necrosis factor (TNF-α; Figure 7B) in DN groups was statistically significantly higher compared to the control groups (*p* < 0.05) and DMT2N0 stage 1 (*p* < 0.05). The results for interferon gamma (IFN-γ; Figure 7C) showed a statistically significant increase in the groups with loss of renal function compared to the controls (DMT2N1 stage 3a, *p* < 0.05, DMT2N2 stage 3b, *p* < 0.05) and DMT2N0 stage 1 (*p* < 0.05). Transforming growth factor-β (TGF-β; Figure 7D) showed statistically significant higher levels in groups with impaired renal function compared to healthy volunteers (DMT2N1 stage 3a, *p* <0.05, DMT2N2 stage 3b, *p* < 0.05) and DMT2N0 stage 1 (*p* < 0.05). Interleukin 10 (IL-10) levels in the DMT2N0 stage 1 group were statistically significant higher compared to DMT2N1 stage 3a (*p* <0.05) and DMT2N2 stage 3b (*p* < 0.05). In groups with impaired kidney function (DMT2N1 stages 3a and DMT2N2 stage 3b), statistically significantly lower levels of IL-10 compared to controls (*p* < 0.05) were observed (Figure 7E).

## 3. Discussion

In recent years, oxidative stress has been among the most studied topics in redox biology, due to the dual role of ROS in numerous physiological and pathophysiological processes in the body. Oxidative stress is defined as a phenomenon involving altered homeostasis that leads to increased production of free radicals in concentrations well above the body’s ability to detoxify (oxidative distress) [12]. The primary purpose of a redox homeostatic system is to allow cells or tissues to respond to environmental challenges by activating adaptive mechanisms involving redox reactions that maintain a delicate balance between reversible or irreversible oxidation of molecular targets, such as transcription factors (factor 2 (Nrf2)) [13]. To be classified as physiologically important signaling molecules, non-radical species, such as H_2_O_2_, or singlet molecular oxygen, must be produced in a tightly programmed and regulated pathway and react with target molecules through specific mechanisms. An example is the role of H_2_O_2_ as a secondary signaling molecule in insulin signaling, proliferation, differentiation, tissue repair, inflammation, circadian rhythm, and aging [14]. At physiological concentrations, H_2_O_2_, or singlet molecular oxygen, performs the function of second messengers, which, together with Nrf2, participate in redox signaling and support the adaptation of intracellular redox homeostasis [13]. At 1–10 nM, H_2_O_2_ participates in redox signaling and regulation of processes at the cellular and molecular level, while its higher concentrations initiate adaptive stress responses (eustress). Concentrations of H_2_O_2_ (>100 nM) are considered supraphysiological and lead to macromolecule damage, referred to as oxidative stress [15]. The primary purpose of a redox homeostatic system is to allow cells or tissues to respond to environmental challenges by activating adaptive mechanisms involving redox reactions that maintain a delicate balance between reversible or irreversible oxidation of molecular targets, such as transcription factors (factor 2 (Nrf2) [13]. Redox regulation of the insulin signaling pathway is initiated by binding the insulin ligand to the insulin receptor (IR), resulting in a cascade of phosphorylation reactions. Many macromolecules involved in insulin signaling, including the insulin receptor, contain cysteine residues whose activity can be changed in response to ROS. The H_2_O_2_ modifies specific cysteine residues in the receptors themselves and significant components of relevant pathways, expressed in reactivation of the action of phosphatases as the dual-specificity phosphatase (PTEN) and protein tyrosine phosphatase (PTP1B), resulting in inhibition of insulin signaling [16].

The complex metabolic environment of diabetes triggers various pathophysiological mechanisms that, depending on the stage of the disease, can increase or decrease the production of NO and its metabolites. Increased levels of NO metabolites in serum have been found in patients with normal or mildly compromised renal function and DN [17,18,19]. Sokolovska et al. [20] found that DM patients without renal impairment had significantly increased levels of NO in serum and decreased levels of nitrite/nitrate (NO_2_^−^/NO_3_^−^). The development and progression of DN is associated with changes in the expression and activity of NOS enzymes and, in particular, eNOS [20]. Experimental and clinical studies highlight that the expression and activity of renal eNOS increase in the initial stage of diabetic nephropathy (DN) and decrease in the chronic diabetic state [21]. Overproduction of NO may contribute to hyperfiltration and microalbuminuria, which is characterized by increased intrarenal NO production mediated primarily by eNOS and nNOS. eNOS is the major NOS enzyme in the renal vasculature, and expression of the enzyme is increased in the early stages of high-glucose-induced renal injury [22,23]. For example, in patients with nephropathy and microalbuminuria, increased expression of eNOS was observed in the glomerular endothelium, along with generalized NO deficiency and fibrosis manifested by either diffuse or nodular changes [24]. These findings are confirmed by the analysis of the results presented in the present study for patients with diabetes and normal renal function and those with moderate and severe renal loss. In type 2 diabetes mellitus patients, increased levels of NO and increased activity of eNOS and iNOS in plasma were observed compared to healthy volunteers. In patients with renal impairment (stages 3a and 3b), the enzymatic activity of eNOS and iNOS was dramatically decreased compared to the control and T2DM groups (Figure 1A–C), which may contribute to the development and progression of diabetic nephropathy [23,24,25].

Hyperglycemia, redox imbalance, endothelial dysfunction, TGF-β activation, etc. (Figure 2B and Figure 7D), lead to reduced production and availability of NO, which underlies the development and progression of diabetes [26,27,28]. In patients with diabetes and neuropathy, in addition to TNF-α, IL-6 production is increased (Figure 7A) [29]. Impaired expression of NOS enzymes may be due to changes in signaling pathways, elevated cytokine levels, and chronic inflammation. It is proposed that eNOS serves as a key mediator in the development and progression of DN, and ROS are strongly involved in this mechanism. Decompensated T2DM diabetes may compromise endothelial function by initiating an imbalance in eNOS expression/activity and reducing the amounts of eNOS substrates/cofactors [30]. eNOS and the pro-inflammatory enzyme iNOS accelerate the conversion of NO into ROS [31]. High levels of ROS and RNS, together with toxic peroxynitrite (ONOO^−^), lead to oxidative modification of proteins, loss of enzyme activity, changes in cellular function, and disruption of cellular homeostasis [32]. The relationship between NO and ROS is bidirectional. Low levels of NO in the endothelium induce the expression of antioxidant genes, while increased levels of ROS downregulate NO production by inhibiting NOS enzymes [33]. In the kidney, elevated ROS levels trap NO, leading to deficiency in both NO and NOS enzymes and generation of ONOO^−^ [34]. As a result, renal endothelial dysfunction, increased vascular resistance, and decreased vasodilation occur [31,32]. OS is directly associated with podocyte damage, increased proteinuria, and the development of tubulointerstitial fibrosis, all of which contribute to the progression of CKD [22,23].

Lipid peroxides (LPx) reflects the impairment of cell membrane integrity and is implicated in various pathological conditions [23,24]. The products of LPx include conjugated dienes and lipid hydroperoxides, thiobarbituric acid reactive substances, gaseous alkanes, and prosta-glandins (F2-PGF) [25,34]. It is known that oxidative damage to lipids and proteins can contribute to insulin resistance and additional complications associated with diabetic nephropathy [35]. MDA binds covalently to lipids and proteins and leads to the formation of cross-links, which disrupts the interaction between insulin and its receptors, reducing the number of insulin-binding sites and contributing to insulin resistance [26]. Increased levels of ROS and LPx were found in patients with diabetic nephropathy, especially in stages 3a and 3b (Figure 2 and Figure 3).

The prostaglandins are considered the most valuable biomarker for lipid peroxidation and is formed in vivo during free radical-induced peroxidation of arachidonic acid [36,37]. On the one hand, increased levels of ROS lead to increased concentrations of prostaglandin compounds; in particular, 8-Iso-PGF [38,39]. On the other hand, high glucose concentrations induce the production of F2-isoprostanes in vascular endothelial smooth muscle cells and contribute to increased formation of F2-PGF in T2DM [40,41]. In Figure 2C, the levels of 8-Iso-PGF increase significantly as the stage of kidney damage progresses (*p* = 0.0001), and they are inversely proportional to eGFR (*r* = −0.92; *p* ≤ 0.00; Figure 3B). This indicates that higher levels of 8-Iso-PGF are associated with more severe renal function loss. The study results presented revealed statistically significant increases in LPx markers (MDA and 8-IsoPGF) in DN patients with mild-to-moderate and moderate-to-severe renal function loss (Figure 2). A positive correlation of 8-Iso-PGF levels and glycated hemoglobin levels (*r* = 0.97, Figure 3B), indicating a relationship between peroxidation and glycemic control. The results presented highlight the need for management and treatment of DN and the importance of OS in lipid peroxidation [28,29].

The interplay between NO, ROS, and the formation of AGEs contributes to the pathogenesis and progression of diabetic nephropathy. Intracellular production of AGE precursors damages cells by modifying protein function, as well as altering the cell surface expression of AGE-modified extracellular matrix components. AGEs accumulate in the plasma and tissue proteins of diabetic patients and are associated with the severity of diabetic complications [42,43]. Elevated AGE levels can induce inflammation and fibrosis in the kidneys [39]. During the formation of AGE-mediated free radicals, reactive carbonyl compounds are generated. These carbonyl compounds are known precursors of carbonyl stress, which further contributes to oxidative damage and cellular dysfunction. Plasma proteins that have been modified by AGE precursors can bind to AGE receptors; particularly, the receptor for advanced glycation end products (RAGE) [44,45]. The AGE–RAGE axis plays a significant role in renal inflammation, fibrosis, and oxidative stress in diabetic nephropathy [46].

Oxidative stress and glycation processes can contribute to the development and progression of diabetic nephropathy, leading to glomerular and tubular damage, impaired filtration, and renal dysfunction [43,44]. Increased PC levels indicate the presence of OS in diabetic nephropathy and reflect the extent of protein oxidation and damage in DN (Figure 4A). In diabetes, chronic hyperglycemia can contribute to the increased formation and accumulation of AGEs (Figure 4B). AGEs can lead to the cross-linking of proteins and the activation of inflammatory pathways, contributing to renal damage and dysfunction [44]. The elevated levels of PC (*p* < 0.05) and AGEs (*p* < 0.05) further increase kidney damage and exacerbate the risk of renal complications in diabetic nephropathy patients.

The development of complications in diabetic nephropathy is associated with high levels of oxidative stress, increased production of ROS and RNS, and disturbances in the process of glycation and glycol oxidation [42]. Figure 6 showed the drastic deficiencies in antioxidant enzyme activity in patients with DN (stages 3a and 3b), which are associated with glycation and glycol oxidation disorders. The reduced SOD activity suggests an impaired ability to counteract oxidative stress in DN. The lower SOD and GPx activity in DN stage 3a and stage 3b indicates a progressive decline in the antioxidant defense system (Figure 6A,C). Increased OS burdens the CAT activity and results in high enzyme activity in DN groups (Figure 6B). This depletion of the antioxidant enzymes contributes to increase OS and may play a crucial role in the pathogenesis and progression of diabetic nephropathy disease [43].

The interplay between oxidative stress, immune dysregulation, and inflammation contributes to the pathogenesis and progression of DN [46]. OS can increase cytokine production via activating the transcription factor nuclear factor kappa B (NFκB), leading to the transcription of genes encoding cytokines. The enhanced macrophage migration induces the release of an inflammatory reaction, which further stimulates ROS production [47,48]. Due to the increased activity of T-lymphocytes, there is an upregulation of Th-1 and Th-2 signaling pathways [49], leading to the depletion of NO levels and NOS enzymes (Figure 1). In a state of hyperglycemia, the activity of nuclear NF-κB increases, resulting in the release of a large number of cytokines, TGF-β, and chemokines. The interplay between oxidative stress, immune dysregulation, and inflammation contributes to the pathogenesis and progression of diabetic nephropathy [25].

Figure 7 shows the significantly higher plasma levels of TNF-α, IFN-γ, TGF-β, and IL-6 in patients with diabetic nephropathy (stage 3a and 3b) and microalbuminuria (Table 1) compared to the levels seen in DMT2N0 patients (Figure 7A–D). High levels of TGF-β1 have been linked to the development of renal hypertrophy in patients with insulin-dependent diabetes mellitus (Figure 7D). This is particularly evident in cases of chronic renal failure, where the degree of deviation in the glomerular filtration rate (eGFR) correlates with the severity of oxidative stress (Figure 2, correlation between MDA and eGFR). Plasma levels of anti-inflammatory IL-10 in patients with microalbuminuria and DN were significantly lower and statistically different from healthy controls and patients with T2DM (Figure 7E). These results confirm the role of inflammatory factors in diabetic kidney disease [25].

An imbalance between pro- and antioxidants in favor of pro-oxidants is at the root of the aging process and the development of many diseases, including diabetes, cardiovascular disease, cancer, and others. It is known that OS and chronic low-grade inflammation contribute to diabetic complications, such as neuropathy, nephropathy, retinopathy, and cardiovascular disease [50]. Type 2 diabetes mellitus is a metabolic disorder characterized by high blood sugar levels resulting from insulin resistance and insufficient insulin production. As the disease progresses, it can lead to complications, including OS and damage due to ROS production. The increased ROS production, such as mitochondrial dysfunction or NADPH oxidase activation, can contribute to the oxidative stress observed in diabetic nephropathy. This oxidative stress can lead to inflammation, endothelial dysfunction, and fibrosis, ultimately resulting in kidney damage [18]. In the context of T2DM, chronic hyperglycemia and insulin resistance contribute to a state of OS, which occurs in a few mechanisms. Elevated glucose levels in the blood can lead to an increased influx of glucose into cells, including mitochondria. This excess glucose metabolism within the mitochondria can overwhelm the respiratory chain and electron transport system, leading to the leakage of electrons and the generation of superoxide radicals [51]. NADPH oxidase is an enzyme that generates ROS as part of the immune response. In the context of T2DM, its activity can be increased, further contributing to ROS production. High blood sugar levels can lead to the formation of AGEs, which are molecules that contribute to oxidative stress and inflammation [52]. As T2DM progresses, due to the toxic effects of the hyperglycemic environment, the compromise of the mitochondrial respiratory chain and NADPH oxidase leads to the production of ROS (mainly O_2_^•−^) in pancreatic β-cells [53] and increased levels of proinflammatory cytokines and inflammatory markers, such as TNF-α anti-inflammatory cytokines, and a decrease in proinflammatory cytokines, e.g., TNF-α, IL-1β, IL-6, IL-18, and C-reactive protein (CRP) [54]. 

Maintaining steady but high levels of oxidants also activates the expression of the proinflammatory cytokines IL-1 and IL-18, which promotes aging and exacerbates cellular damage. Indeed, with advancing age, chronic inflammatory stress is observed as a result of immune system activity, which may be due to systemic oxidative inflammatory stress. It is known that chronic inflammation and the increased and uncontrolled production of free radicals can trigger an inflammatory response due to the reply of immune cells (monocytes, macrophages, and lymphocytes) to pathogens, leading to increased production of ROS, proinflammatory mediators, and cell damage accompanied by overproduction of ROS. At the same time, the body’s inflammatory response initiates additional ROS generation, for example, mitochondrial ROS (mROS) [55]. Oxidative stress-induced mitochondrial damage and destruction of mitochondria results in a lack of energy needed for glucose uptake, further exacerbating insulin resistance. Thus, high levels of ROS and proinflammatory cytokines inhibit mitochondrial function and contribute to pancreatic β-cell apoptosis and insulin resistance, ultimately leading to the development of T2DM. Prolonged hyperglycemia further induces the production of ROS and RNS (e.g., O_2_^•−^, H_2_O_2_, NO), leading to a vicious cycle between forming and accumulating free radicals and systemic inflammation [56].

## 4. Materials and Methods

### 4.1. Chemicals

All reagents were analytical grade (N-tert-butyl-α-phenylnitrone, PBN (B7263; CAS no. 3376-24-7), carboxy-PTIO potassium salt (C221; CAS no. 148819-94-7), dimethyl sulfoxide DMSO (472301 CAS no. 472301), 2-thiobarbituric acid (T5500; CAS no. 504-17-6)) and were purchased from Merck, Sigma-Aldrich, Sofia, Bulgaria EAD.

### 4.2. Subjects

This study included 111 patients with diabetes mellitus (DMT2) (male: female ratio: 1:1.3). The wide range of variables associated with the demographic characteristics reflected the broad spectrum of our DM population. About half of the diabetic patients (58.1%) were obese. The patients were divided into three groups according to the stage of diabetic nephropathy. The results were compared with 36 age-matched healthy volunteers. The venous blood of fasting patients and healthy volunteers was collected in the morning after overnight fasting for lipid profile analysis. All collected samples for the electron paramagnetic resonance (EPR) study were studied immediately and then frozen at −80 °C for the ELISA test. Fasting plasma glucose (FPG) concentrations and glycated hemoglobin (HbA1c%), representing an average measure of glycemic exposure over time, were measured as parameters for glycemic control. The experiment was conducted following the ethical standards of the Medical Faculty and Hospital Research Committee and the Helsinki Declaration of 1964 and its later changes or comparable ethical standards. Patients were diagnosed with T2DM and diabetes with a complication of diabetic nephropathy using the GFR and UAE values (Table 1).

At the time of inclusion in the study, 54 (58%) DMT2N0 patients had poor disease compensation, defined as HbA1c > 7% and fasting blood glucose values >6.1 mmol/L. Regarding the therapy carried out, patients from the T2DMN0 group (*n* = 21), diabetics with good glycemic control, took oral hypoglycemic agents (sulfonylureas and biguanides). Diabetic patients with nephropathy (DMT2N1, *n* = 43) with mild-to-moderate loss of renal function (eGFR stage 3a), *n* = 20, were on sulfonylureas and biguanides, and the remaining *n* = 23 were on sulfonylureas and basal insulin.

All patients (DN, *n* = 47) with moderate-to-severe loss of renal function and eGFR stage 3b were on basal-bolus insulin therapy (rapid insulin preprandial and evening dose—basal insulin). The comparison was made with 36 healthy volunteers. Only 7 (7.4%) of the controls had slightly increased body weight (BMI 31–33). Participants were matched for gender.

### 4.3. Electron Paramagnetic Resonance (EPR) Study

All EPR measurements were performed at room temperature on a Bruker BioSpin GmbH, Ettlingen, Germany, equipped with a standard resonator. All EPR experiments were carried out in triplicate and repeated thrice. Spectral processing was performed using Bruker WIN-EPR and Simfonia software, 2009.

#### 4.3.1. Evaluation of the ROS Product Levels

The levels of ROS were determined following Shi et al. [57] with some modification. To investigate in real time the formation of ROS in the sera of patients and controls, ex vivo EPR spectroscopy was used combined with N-tert-butyl-alpha-phenylnitrone (PBN) as a spin-trapping agent. PBN, upon reaction with unstable radicals, such as ROS, forms relatively stable spin adducts that can be subsequently detected by EPR spectroscopy. Briefly, to 100 μL plasma was added 900 μL 50 mM PBN dissolved in dimethyl sulfoxide (DMSO), and, after centrifugation at 4000× *g* rpm for 10 min at 4 °C, the EPR spectra were immediately recorded in the supernatant. The levels of ROS products were calculated as double-integrated plots of EPR spectra and results were expressed in arbitrary units (a. u.).

The EPR settings were as follows: 3503.73 G center field, 20.00 mW microwave power, 5 G modulation amplitude, 50 G sweep width, 1 × 10^5^ gain, 81.92 ms time constant, 125.95 s sweep time, 5 scans per sample.

#### 4.3.2. Evaluation of the •NO Radical Levels

Based on the methods published by Yoshioka et al. [58] and Yokoyama et al. [59], we developed and adapted the EPR method for estimation of •NO radical levels. Briefly, to a 50 μM solution of carboxy 2-(4-carboxyphenyl)-4,4,5,5-tetramethyl was added imidazoline-1-oxyl-3-oxide potassium salt (carboxy PTIO.K) dissolved in a mixture of 50 mMTris (pH 7.5) and DMSO in a ratio 9:1.

To 100 μL plasma was added 900 μL Tris buffer dissolved in DMSO (9:1). After that, the mixture was centrifuged at 4000× *g* rpm for 10 min at 4 °C. Then, 100 μL of sample and 100 μL 50 mM solution of carboxy PTIO.K were mixed and EPR spectra of spin adducts formed between the spin trap carboxy PTIO.K and generated •NO radicals were recorded. The levels of •NO radicals were calculated as double-integrated plots of EPR spectra and results were expressed in a.u.

The EPR settings were as follows: 3505 G centerfield, 6.42 mW microwave power, 5 G modulation amplitude, 75 G sweep width, 2.5 × 10^2^ gain, 40.96 ms time constant, 60.42 s sweep time, 1 scan per sample.

### 4.4. Enzyme-Linked Immunosorbent Assay

All markers of oxidative stress were measured with ELISA kits following the manufacturer’s instructions. The ELISA kits were as follows: Human eNOS (ab241149); Human iNOS ELISA Kit (ab253217); 8 isoprostane ELISA Kit (ab175819); Human Protein Carbonyl ELISA Kit (ab238536); Human AGEs levels ELISA Kit (ab238539); Human IL-6 ELISA Kit (ab178013); Human IL-10 ELISA Kit (ab185986); Human TNF alpha ELISA Kit (ab181421); Human IFN gamma High Sensitivity ELISA Kit (ab236895); Human TGF beta 1 ELISA Kit (ab100647).

### 4.5. Spectrophotometric Determination of Lipid Peroxidation Products—Malondialdehyde (MDA)

The total amount of lipid peroxidation products in the plasma of healthy volunteers and patients was estimated using the thiobarbituric acid (TBA) method described by Plaser et al. [60], which measures the malondialdehyde (MDA) reactive products at 532 nm; results were expressed in μmol/L.

### 4.6. Statistical Analysis

Statistical analysis was performed with Statistica 8, StaSoft, Inc. (Madrid, Spain), and the results were expressed as means ± S.E. All data were expressed as means ± SE and obtained by one-way ANOVA, and in the LSD post hoc test, *p* > 0.05 was considered statistically significant. To define which groups were different from each other, LSD post hoc tests were used.

## 5. Conclusions

The role of oxidative stress in various diseases is gaining attention from clinicians, as it has been implicated in the pathogenesis of many pathological conditions. When there is excessive ROS production in the body, it can disrupt the balance of homeostasis and lead to pathological symptoms. ROS can cause damage to cells and tissues, as well as induce changes in gene expression, further contributing to disease progression. In the context of the current study, compromised oxidative status can be suggested to underlie renal function impairment in patients with decompensated type 2 diabetes mellitus. Patients with impaired renal function, characterized by a glomerular filtration rate below 60 mL/min but above 30 mL/min, are unable to adequately compensate for OS. This inability to maintain a proper redox balance leads to the initiation of chronic inflammation, which is believed to play a significant role in the development and progression of diabetic nephropathy.

Therefore, identifying oxidative stress biomarkers and assessing inflammation can contribute to monitoring the overlapping complications commonly encountered in diabetes. By understanding the interplay between oxidative stress, redox regulation, and inflammation, we can gain insights into the mechanisms underlying diabetic nephropathy and other complications associated with diabetes. Monitoring oxidative stress biomarkers and inflammation levels can provide valuable information for the management and treatment of these conditions. By targeting and addressing oxidative stress and inflammation, clinicians can potentially mitigate the progression of diabetic nephropathy and improve patient outcomes.

## Figures and Tables

**Figure 1 ijms-24-13541-f001:**
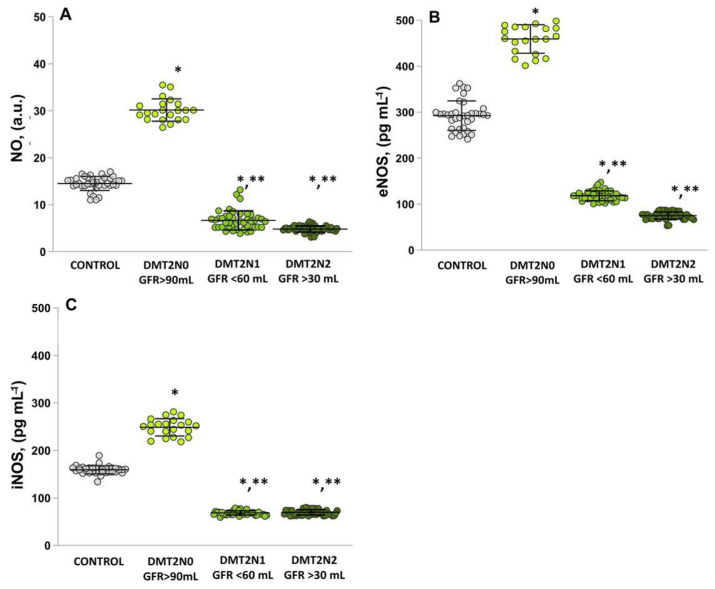
Present the levels of nitric oxide (NO), eNOS and iNOS in plasma samples. (**A**) NO: (1) healthy controls; (2) DMT2N0 patients with normal kidney function (eGFR ≥ 90 mL/min/1.73 m^2^, stage 1); (3) DMT2N1 with mild-to-moderate renal function loss (eGFR < 60 mL/min/1.73 m^2^, stage 3a); (4) DMT2N2 with moderate-to-severe renal function loss (eGFR > 30 mL/min/1.73 m^2^, stage 3b). (**B**) eNOS: (1) healthy controls; (2) DMT2N0 patients with normal kidney function (eGFR ≥ 90 mL/min/1.73 m^2^, stage 1); (3) DMT2N1 with mild-to-moderate renal function loss (eGFR < 60 mL/min/1.73 m^2^, stage 3a); (4) DMT2N2 with moderate-to-severe renal function loss (eGFR > 30 mL/min/1.73 m^2^, stage 3b). (**C**) iNOS: (1) healthy controls; (2) DMT2N0 patients with normal kidney function (eGFR ≥ 90 mL/min/1.73 m^2^, stage 1); (3) DMT2N1 with mild-to-moderate renal function loss (eGFR < 60 mL/min/1.73 m^2^, stage 3a); (4) DMT2N2 with moderate-to-severe renal function loss (eGFR > 30 mL/min/1.73 m^2^, stage 3b). LSD post hoc test; * *p* < 0.05 vs. control group; ** *p* < 0.05 vs. DMT2N0 group (stage 1).

**Figure 2 ijms-24-13541-f002:**
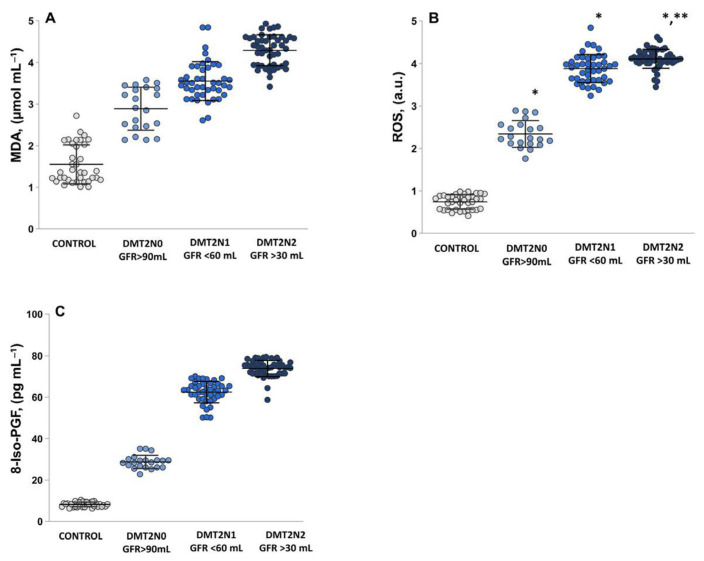
Displays the levels of oxidative stress markers presented as MD, ROS production and 8-Iso-PGF. (**A**) MDA levels—controls; DMT2N0 stage 1; DMT2N1 stage 3a; DMT2N2 stage 3b. (**B**) ROS production—controls; DMT2N0 stage 1; DMT2N1 stage 3a; DMT2N2 stage 3b. (**C**) 8-Iso-PGF—controls; DMT2N0 stage 1; DMT2N1 stage 3a; DMT2N2 stage 3b. LSD post hoc test; (*) *p* < 0.05 vs. control; (**) *p* < 0.05 vs. DMT2N0 stage 1.

**Figure 3 ijms-24-13541-f003:**
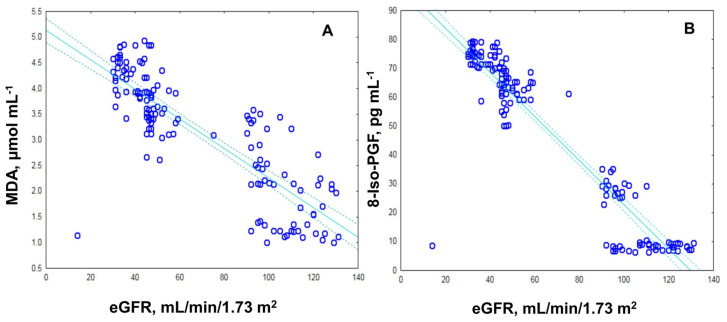
Correlation graphs presented as individual measurements. (**A**) correlation between MDA and eGFR; (**B**) correlation between 8-IsoPGF and eGFR; (**C**) correlation between MDA and UAE; (**D**) correlation between 8-IsoPGF and UAE.

**Figure 4 ijms-24-13541-f004:**
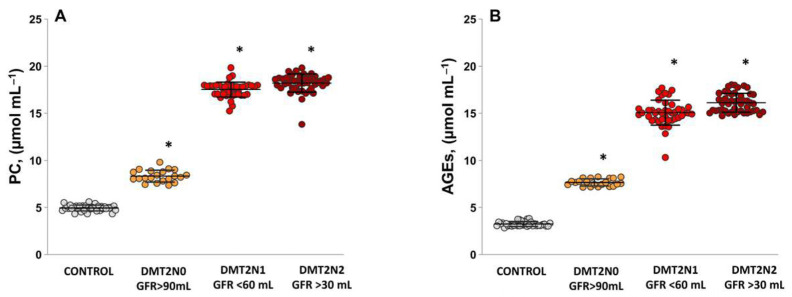
Levels of protein carbonyls and advanced glycation end products. (**A**) PC levels—controls; DMT2N0 stage 1; DMT2N1 stage 3a; DMT2N2 stage 3b. (**B**) AGEs levels—controls; DMT2N0 stage 1; DMT2N1 stage 3a; DMT2N2 stage 3b. * *p* < 0.05 vs. control.

**Figure 5 ijms-24-13541-f005:**
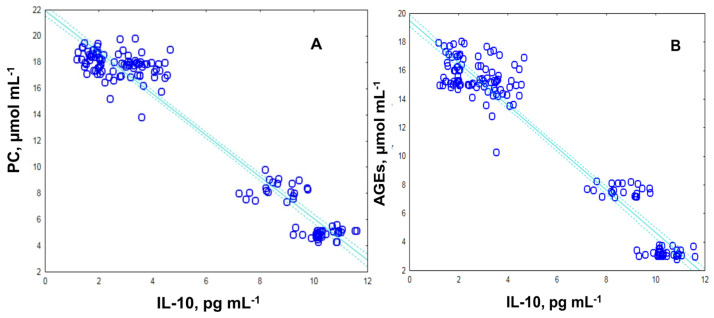
Scatter plot between plasma levels of (**A**) PC and IL-10 and (**B**) AGEs and IL-10.

**Figure 6 ijms-24-13541-f006:**
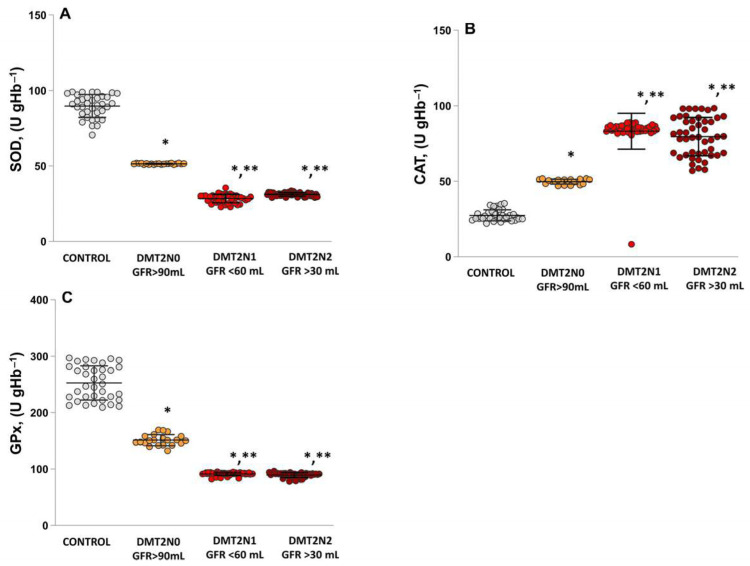
Activity of antioxidant enzymes –SOD, CAT and GPx. (**A**) SOD activity—controls; DMT2N0 stage 1; DMT2N1 stage 3a; DMT2N0 stage 3b. (**B**) CAT activity—controls; DMT2N0 stage 1; DMT2N1 stage 3a; DMT2N2 stage 3b. (**C**) GPx activity—controls; DMT2N0 stage 1; DMT2N1 stage 3a; DMT2N2, GFR stage 3b. LSD post hoc test; (*) *p* < 0.05 vs. control; (**) *p* < 0.05 vs. DMT2N0 stage 1.

**Figure 7 ijms-24-13541-f007:**
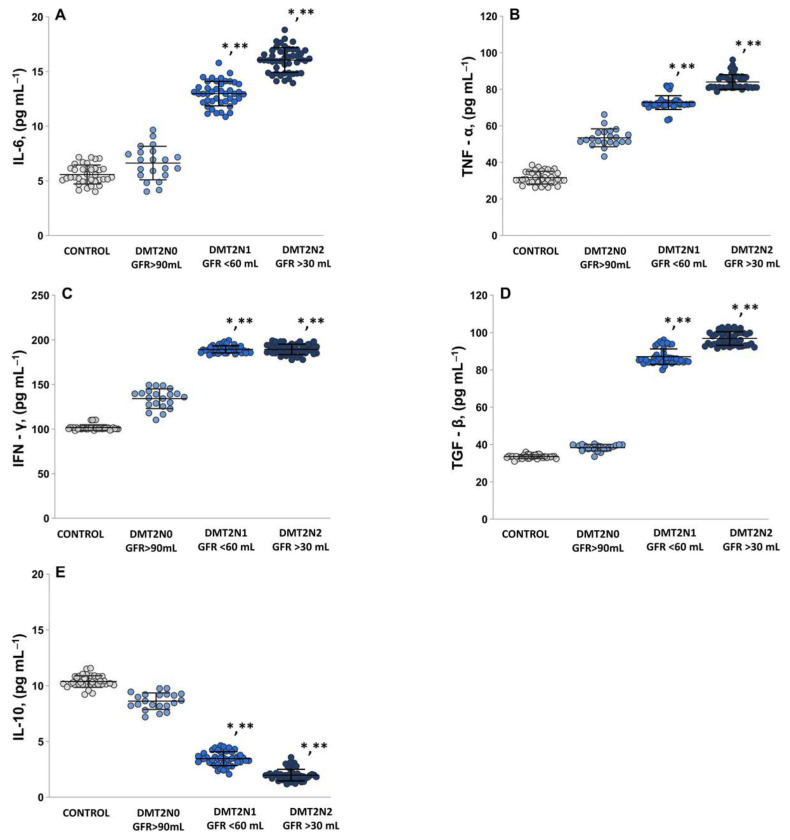
Pro-inflammatory cytokine levels: (**A**) IL-6; (**B**) TNF-α; (**C**) IFN-γ; (**D**) TGF-β; (**E**) IL-10. LSD post hoc test, * *p* < 0.05 vs. control; ** *p* < 0.05 vs. DMT2N0 stage 1.

**Table 1 ijms-24-13541-t001:** Clinical and laboratory data for patients depending on the levels of eGFR (mL/min) and UAE (mg/day). The results were compared with the data for healthy volunteers (controls). Data are shown for patients with T2DM with normal renal function (eGFR ≥ 90 mL/min/1.73 m^2^; stage 1), patients with complications of diabetic nephropathy with mild-to-moderate loss of renal function (eGFR 59 to 45 (>60) mL/min/1.73 m^2^; stage 3a), and patients with diabetic nephropathy with moderate-to-severe loss of renal function (eGFR 44 to 30 (<30) mL/min/1.73 m^2^; stage 3b).

Variables	Controls (*n* = 36)	DMT2N0 with Normal Renal Function (*n* = 21), Stage 1	DMT2N1 with Mild-to-Moderate Renal Failure (*n* = 43), Stage 3a	DMT2N2 with Moderate-to-Severe Renal Failure (*n* = 47), Stage 3b
Age	41.25 ± 7.78	51.27 ± 6.07	54.41 ± 5.74	56.68 ± 8.91
Sex (M/F)	14 M/24 F	10 M/11 F	10 M/13 F	14 M/10 F
Disease duration	-	10.7 ± 7.3	12.7 ± 8.6	14.8 ± 6.1
BMI (kg/m^2^), mean ± SD	26.07 ± 3.52	32.11 ± 5.44	33.56 ± 6.21	34.76 ± 6.14
Blood sugar (mmol/L), mean ± SD	4.98 ± 0.32	8.05 ± 5.11	9.55 ± 4.89	9.82 ± 5.41
HbA1_C_%	5.69 ± 0.44	6.37 ± 0.85	9.87 ± 1.27	9.99 ± 1.98
GFR (mL/min/1.73 m^2^)	110.08 ± 9.32	95.66 ± 5.60	49.83 ± 2.57	36.95 ±4.50
UAE (mg/day)	1.20 ± 035	21.80 ± 1.7	213.55 ± 6.85	651.08 ± 8.92
FBC (mmol/L)	5.31 ± 0.28	6.43 ± 0.89	12.72 ± 5.15	5.47± 0.6
Cholesterol (mmol/L)	4.17 ± 0.3	5.01 ± 0.7	5.47 ± 0.6	5.47 ± 0.6
Triglycerides (mmol/L)	1.31 ± 0.15	2.13 ± 0.2	2.65 ± 0.2	3.01
HDL (mmol/L)	0.91 ± 0.019	1.19 ± 0.04	1.32 ± 0.05	1.33 ± 0.01
LDL (mmol/L)	2.03 ± 0.11	2.72 ± 0.24	2.96 ± 0.2	3.2 ± 0.5

## Data Availability

The data presented in this study are available on request from the corresponding author.

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
