# Peer review of "Biomarkers of Oxidative Stress in Diabetes Mellitus with Diabetic Nephropathy Complications"

_ijms, 2023, doi:10.3390/ijms241713541_

Round 1

Reviewer 1 Report

The authors investigate and compare biomarkers of oxidative stress (OS) and the activity of antioxidant enzymes in the plasma of patients with different stages of diabetic nephropathy (DN). The manuscript is well-written and easy to read. The biomarkers are very important for the follow-up of diseases. However, the samples from which the biomarkers have to be assessed must be well described. 

1-We would like to understand why the authors chose the plasma sample instead of serum.

2-The collection of plasma must be described. The duration between blood sample collection and plasma preparation. 

3- The statistics analyses why the authors only used a t-test, the Anova multi-comparisons should give more correlations of biomarkers in the different phases of T2DM.

4-Please, define some abbreviations before using them.

5- figure 3 legend, line 234: AEGs or AGEs? line 279: CAT or SAT?

Author Response

Dear Reviewer,

Thank you very much for helping us improve our manuscript.

All corrections are highlighted in yellow

The authors investigate and compare biomarkers of oxidative stress (OS) and the activity of antioxidant enzymes in the plasma of patients with different stages of diabetic nephropathy (DN). The manuscript is well-written and easy to read. The biomarkers are very important for the follow-up of diseases. However, the samples from which the biomarkers have to be assessed must be well described. 

Point 1

We would like to understand why the authors chose the plasma sample instead of the serum.

Response 1: The results provided by us are part of a larger project in which, after an initial established oxidative status of the patients in the examined groups, a special diet with chokeberry (Aronia), wild garlic (Allium vineale), rosemary (Salvia rosmarinus), mint (Mentha ), basil (Ocimum basilicum) combined with yogurt from donkey and goat.

According to our hypothesis, poor T2DM control and persistent hyperglycemia, as well as glucose variations, potentiate the onset and progression of diabetic nephropathy. Microalbuminuria, as a reversible stage of initial diabetic nephropathy, could be influenced both by factors established so far in practice - excellent control of diabetes, arterial blood pressure, early inclusion of ACE-inhibitors/ARBs, and by inclusion in the diet of appropriate antioxidants. Urine protein levels will therefore be tested and compared to plasma levels. Also, the spectrophotometric method for the determination of MDA described by us works in plasma.

Point 2: The collection of plasma must be described. The duration between blood sample collection and plasma preparation. 

Response 2: The venous blood of fasting patients and healthy volunteers was collected in the morning after overnight fasting for lipid profile analysis. All collected samples for the Electron paramagnetic resonance (EPR) study were studied immediately, and after that were frozen at -80°C for the ELISA test. Fasting plasma glucose concentrations (FPG) and glycated hemoglobin (HbA1c %), representing an average measure of glycemic exposure over time, were measured as parameters for glycemic control.

Point 3: The statistics analyze why the authors only used a t-test, the Anova multi-comparisons should give more correlations of biomarkers in the different phases of T2DM.

Response 3: We add LSD post hoc test, correlations, and new figures

Point 4: Please, define some abbreviations before using them.

Response 4: Done

Point 5: figure 3 legend, line 234: AEGs or AGEs? line 279: CAT or SAT?

Response 5: Done

Reviewer 2 Report

The study of Goycheva and collegues is potentially interesting, however is merely descriptive by measuring some blood parameters from patients with different stages of DN. The paper need a complete and susbtantial revision to be considered for publication. Also, it has been known for decades that oxidative stress and DN progression are related, and that drugs inhibiting PDE activity or stimulating cGMP production are beneficial.

Major concerns:

1) Introduction is too long

2) The reslts section contains significant amount of discussion, which should be entirely moved to the discussion section. One sentence explanation of the actual result should be the maximum within this section. eg. Lines 142-148, 185-197, 240-260 etc should be moved into discussion!

3) The results graphs should be changed to scatter plots to show individual data points! Delete the "plasma drop" as all samples are plasma, maked no sense.

4) The two-star significance referred to as "p<0.05 to OTA" needs explanation - what is OTA? No such group exists.

5) Line 167: no correlation graph is shown, please include

6) Line 178-180: in case of 4 groups pair-wise t-test is not allowed, these sentences should be omitted. Permorm ANOVA with appropriate post-hoc test to compare each group to each group.

7) Discussion is overwhelmed compared to the relative small amount of data presented.

8) What was the rationale to measure intracellular enzyme concentrations in the blood? Please include strong evidences from the literature that plasma eNOS and iNOS levels are relevant. Also, measurement of NO level from blood rise questions, as it is known to have extremely short half-life (!) but its signaling targets would be better alternative (plasma cGMP, urine cGMP, SGC-PKG-PDE axis components). 

9) In Table 1, exact values for urinary albumin excretion should be given. Having an AUE of 50 or 290 makes shows a huge difference in kidney damage, still the authors include both into the "30-300" group. Moreover, exact UAE levels could be correlated with the measured parameters.

10) The methods section is very superficial, and impedes reproducibility of the study. Vendors, catalogue numbers should be described for each kit or reagent used!

Author Response

Response to Reviewer Comments

Dear Reviewer,

Thank you very much for helping us improve our manuscript.

All corrections are colored red.

The study of Goycheva and colleagues is potentially interesting, however, is merely descriptive by measuring some blood parameters from patients with different stages of DN. The paper needs a complete and substantial revision to be considered for publication. Also, it has been known for decades that oxidative stress and DN progression are related and that drugs inhibiting PDE activity or stimulating cGMP production are beneficial.

Major concerns:

Point 1: The introduction is too long

Response 1: The introduction was shortened.

Point 2: The results section contains a significant amount of discussion, which should be entirely moved to the discussion section. One sentence explanation of the actual result should be the maximum within this section. eg. Lines 142-148, 185-197, 240-260 etc should be moved into discussion!

Response 2: Done

Point 3: The results graphs should be changed to scatter plots to show individual data points! Delete the "plasma drop" as all samples are plasma, makes no sense.

Response 3: Done

Point 4: The two-star significance referred to as "p<0.05 to OTA" needs explanation - what is OTA? No such group exists.

Response 4: Corrected

Point 5: Line 167: no correlation graph is shown, please include

Response 5: Included

Point 6: Line 178-180: in case of 4 groups pair-wise t-test is not allowed, these sentences should be omitted. Permorm ANOVA with appropriate post-hoc test to compare each group to each group.

Response 6: Done

Point 7: Discussion is overwhelmed compared to the relative small amount of data presented.

Response 7: We rewrote a discussion

Point 8: What was the rationale to measure intracellular enzyme concentrations in the blood? Please include strong evidences from the literature that plasma eNOS and iNOS levels are relevant. Also, measurement of NO level from blood rise questions, as it is known to have extremely short half-life (!) but its signaling targets would be better alternative (plasma cGMP, urine cGMP, SGC-PKG-PDE axis components). 

Response 8: The complex metabolic environment in diabetes triggers various pathophysiological mechanisms that, depending on the stage of the disease, can increase or decrease the production of NO and its metabolites. Increased levels of NO metabolites in serum have been found in patients with normal or mildly compromised renal function and DN [9-11]. Sokolovska et al., [12] found that DM patients without renal impairment had significantly increased levels of NO in serum and decreased levels of nitrite/nitrate (NO2‾/NO3‾). The development and progression of DN are associated with changes in the expression and activity of NOS enzymes and in particular eNOS [12]. Experimental and clinical studies highlight that the expression and activity of renal eNOS increase in the initial stage of Diabetic nephropathy DN and decrease in the chronic diabetic state [13]. Overproduction of NO may contribute to hyperfiltration and microalbuminuria, which is characterized by increased intrarenal NO production mediated primarily by eNOS and nNOS. eNOS is the major NOS enzyme in the renal vasculature, and expression of the enzyme is increased in the early stages of high-glucose-induced renal injury [14,15]. For example, in patients with DN nephropathy and microalbuminuria, increased expression of eNOS was observed in the glomerular endothelium, generalized NO deficiency, and fibrosis manifested by either diffuse or nodular changes [16]. These findings are confirmed by the analysis of the results presented in the present study in patients with diabetes and normal renal function and in those with moderate and severe renal loss. In T2DM, increased levels of NO and increased activity of eNOS and iNOS in plasma are observed compared to healthy volunteers. In patients with renal impairment (Stages 3a and 3b), the enzymatic activity of eNOS and iNOS is dramatically decreased compared to control and T2DM (Figure 1 A-C), which may contribute to the development and progression of diabetic nephropathy [15-17].

Point 9: In Table 1, exact values for urinary albumin excretion should be given. Having an AUE of 50 or 290 makes shows a huge difference in kidney damage, still, the authors include both in the "30-300" group. Moreover, exact UAE levels could be correlated with the measured parameters.

Response 9: Done

Point 10: The methods section is very superficial, and impedes reproducibility of the study. Vendors, catalogue numbers should be described for each kit or reagent used!

 Response 10: Done

Round 2

Reviewer 2 Report

In the revised version, the introduction section and figures have been substantially improved. The results section is has been improved to some extent as well, but everal problems still exist, that need corrections.

Here, data being shown on a figure should not be included in the text (avoid redundancies!). (Lines 87-89, 92-93, 150-165, 204-228).

As indicated previously, the results section should not include too much discussion-like explanatory sentences. Lines 95-99 are pure discussion, not belonging to results. Please move these parts to discussion or delete.

Stating a p=0.00 through the whole manuscript text does not have statistically any sense and are invalid (!). Please indicate valid, exact p values (eg. p=0.0028). 

The used statistical method should be indicated in the figure legends rather then repeatedly throughout the main text. Please correct.

In Fig.3 and Fig.5 the graph axis should include dimension of measured parameter, not just IL-10 for example.

In the methods section, supplier of some reagents are still missing (eg., supplier and catalogue numbers of PBN, PTIO.K, etc). Please correct.

There are several parts highlighted in yellow. Please correct.

Overall English is fine. But check and correct typos (eg. ITF-gamma in line 210; Germane in line 407; plsma line 431, etc).

Author Response

Dear Reviewer,

Thank you very much for helping us improve our manuscript.

In the revised version, the introduction section and figures have been substantially improved. The results section is has been improved to some extent as well, but everal problems still exist, that need corrections.

Point 1: Here, data being shown on a figure should not be included in the text (avoid redundancies!). (Lines 87-89, 92-93, 150-165, 204-228).

Response 1: Done

Point 2: As indicated previously, the results section should not include too much discussion-like explanatory sentences. Lines 95-99 are pure discussion, not belonging to results. Please move these parts to discussion or delete.

Response 2: Done

Point 3: Stating a p=0.00 through the whole manuscript text does not have statistically any sense and are invalid (!). Please indicate valid, exact p values (eg. p=0.0028).

Response 3: Done

Point 4: The used statistical method should be indicated in the figure legends rather then repeatedly throughout the main text. Please correct.

Response 4: Done

Point 5: In Fig.3 and Fig.5 the graph axis should include dimension of measured parameter, not just IL-10 for example.

Response 5: Done

Point 6: In the methods section, supplier of some reagents are still missing (eg., supplier and catalogue numbers of PBN, PTIO.K, etc). Please correct.

Response 6: Done

Point 7: There are several parts highlighted in yellow. Please correct.

Response 7: Done

Comments on the Quality of English Language

Point 8: Overall English is fine. But check and correct typos (eg. ITF-gamma in line 210; Germane in line 407; plsma line 431, etc)

Response 8: Done
